# Towards a Mechanistic Interpretation of Multi-Step Reasoning Capabilities of Language Models

**Yifan Hou[1], Jiaoda Li[1], Yu Fei[2], Alessandro Stolfo[1], Wangchunshu Zhou[3], Guangtao Zeng[4], Antoine Bosselut[5], Mrinmaya Sachan[1]**

[1]ETH Zürich, [2]UC Irvine, [3]AIWaves, [4]SUTD, [5]EPFL

[1]{yifan.hou, jiaoda.li, alessandro.stolfo, mrinmaya.sachan}@inf.ethz.ch, [2]yu.fei@uci.edu,
[3]chunshu@aiwaves.cn, [4]guangtao_zeng@mymail.sutd.edu.sg, [5]antoine.bosselut@epfl.ch

## Abstract

Recent work has shown that language models (LMs) have strong multi-step (i.e., procedural) reasoning capabilities. However, it is unclear whether LMs perform these tasks by cheating with answers memorized from pretraining corpus, or, via a multi-step reasoning mechanism. In this paper, we try to answer this question by exploring a mechanistic interpretation of LMs for multi-step reasoning tasks. Concretely, we hypothesize that the LM implicitly embeds a reasoning tree resembling the correct reasoning process within it. We test this hypothesis by introducing a new probing approach (called `MechanisticProbe`) that recovers the reasoning tree from the model's attention patterns. We use our probe to analyze two LMs: GPT-2 on a synthetic task ($k$-th smallest element), and LLaMA on two simple language-based reasoning tasks (ProofWriter & AI2 Reasoning Challenge). We show that `MechanisticProbe` is able to detect the information of the reasoning tree from the model's attentions for most examples, suggesting that the LM indeed is going through a process of multi-step reasoning within its architecture in many cases.[1]

## 1 Introduction

Large language models (LMs) have shown impressive capabilities of solving complex reasoning problems (Brown et al., 2020; Touvron et al., 2023). Yet, what is the underlying "thought process" of these models is still unclear (Figure 1). Do they *cheat with shortcuts memorized from pretraining corpus* (Carlini et al., 2022; Razeghi et al., 2022; Tang et al., 2023)? Or, do they *follow a rigorous reasoning process and solve the problem procedurally* (Wei et al., 2022; Kojima et al., 2022)? Answering this question is not only critical to our understanding of these models but is also critical for the development of next-generation faithful

---

[1]Our code, as well as analysis results, are available at https://github.com/yifan-h/MechanisticProbe.

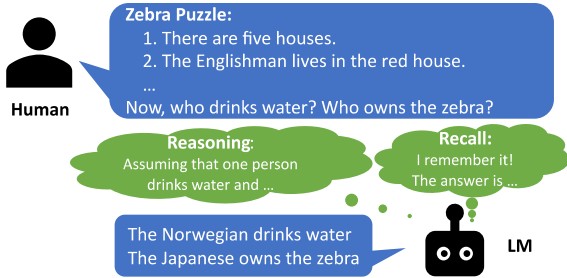

Figure 1: An example showing how LMs solve reasoning tasks. It is unclear if LMs produce the answer by reasoning or by recalling memorized information.

language-based reasoners (Creswell and Shanahan, 2022; Creswell et al., 2022; Chen et al., 2023).

A recent line of work tests the behavior of LMs by designing input-output reasoning examples (Zhang et al., 2023; Dziri et al., 2023). However, it is expensive and challenging to construct such high-quality examples, making it hard to generalize these analyses to other tasks/models. Another line of work, *mechanistic interpretability* (Merullo et al., 2023; Wu et al., 2023; Nanda et al., 2023; Stolfo et al., 2023; Bayazit et al., 2023), directly analyzes the parameters of LMs, which can be easily extended to different tasks. Inspired by recent work (Abnar and Zuidema, 2020; Voita et al., 2019; Manning et al., 2020; Murty et al., 2023) that uses attention patterns for linguistic phenomena prediction, we propose an attention-based mechanistic interpretation to expose how LMs perform multi-step reasoning tasks (Dong et al., 2021).

We assume that the reasoning process for answering a multi-step reasoning question can be represented as a reasoning tree (Figure 2). Then, we investigate if the LM implicitly infers such a tree when answering the question. To achieve this, we designed a probe model, `MechanisticProbe`, that recovers the reasoning tree from the LM's attention patterns. To simplify the probing problem and gain a more fine-grained understanding, we decompose the problem of discovering reasoning trees into two

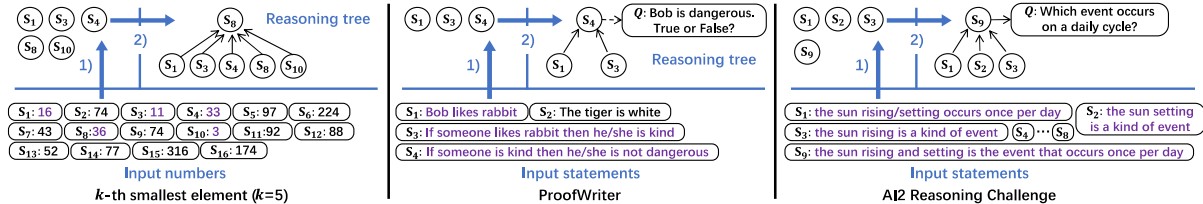

Figure 2: Illustration of our MechanisticProbe with one example from each of the three reasoning tasks considered in this work. We are given a number of input statements: $\mathcal{S} = \{S_1, S_2, ...,\}$ and a question: $Q$. MechanisticProbe recovers the reasoning tree describing the ground-truth reasoning process to answer the question. MechanisticProbe works in two stages: in the first stage, MechanisticProbe detects if the LM can select the set of useful statements required in reasoning, and then in the second stage, MechanisticProbe detects if the LM can predict the reasoning tree given the useful statements.

subproblems: 1) identifying the necessary nodes in the reasoning tree; 2) inferring the heights of identified nodes. We design our probe using two simple non-parametric classifiers for the two subproblems. Achieving high probing scores indicates that the LM captures the reasoning tree well.

We conduct experiments with GPT-2 (Radford et al., 2019) on a synthetic task (finding the $k$-th smallest number in a sequence of numbers) and with LLaMA (Touvron et al., 2023) on two natural language reasoning tasks: ProofWriter (Tafjord et al., 2021) and AI2 Reasoning Challenge (i.e., ARC: Clark et al., 2018). For most examples, we successfully detect reasoning trees from attentions of (finetuned) GPT-2 and (few-shot & finetuned) LLaMA using MechanisticProbe. We also observe that LMs find useful statements immediately at the bottom layers and then do the subsequent reasoning step by step (§4).

To validate the influence on the LM's predictions of the attention mechanisms that we identify, we conduct additional analyses. First, we prune the attention heads identified by MechanisticProbe and observe a significant accuracy degradation (§5). Then, we investigate the correlation between our probing scores and the LM's performance and robustness (§6). Our findings suggest that LMs exhibit better prediction accuracy and tolerance to noise on examples with higher probing scores. Such observations highlight the significance of accurately capturing the reasoning process for the efficacy and robustness of LMs.

## 2 Reasoning with LM

In this section, we formalize the reasoning task and introduce the three tasks used in our analysis: $k$-th smallest element, ProofWriter, and ARC.

### 2.1 Reasoning Formulation

In our work, the LM is asked to answer a question $Q$ given a set of statements denoted by $\mathcal{S} = \{S_1, S_2, ...\}$. Some of these statements may not be useful for answering $Q$. To obtain the answer, the LM should perform reasoning using the statements in multiple steps. We assume that this process can be represented by a reasoning tree $G$. We provide specific examples in Figure 2 for the three reasoning tasks used in our analysis.[2] This is a very broad formulation and includes settings such as theorem proving (Loveland, 1980) where the statements could be facts or rules. In our analyses, we study the tasks described below.

### 2.2 Reasoning Tasks

$k$-**th smallest element.** In this task, given a list of $m$ numbers ($m = 16$ by default) in any order, the LM is asked to predict (i.e., generate) the $k$-th smallest number in the list. For simplicity, we only consider numbers that can be encoded as one token with GPT-2's tokenizer. We select $m$ numbers randomly among them to construct the input number list. The reasoning trees have a depth of 1 (Figure 2 left). The root node is the $k$-th smallest number and the leaf nodes are top-$k$ numbers.

For this task, we select **GPT-2** (Radford et al., 2019) as the LM for the analysis. We randomly generate training data to finetune GPT-2 and ensure that the test accuracy on the reasoning task is larger than $90\%$. For each $k$, we finetune an independent GPT-2 model. More details about finetuning (e.g., hyperparameters) are in Appendix B.1.

**ProofWriter.** The ProofWriter dataset (Tafjord et al., 2021) contains theorem-proving problems.

---

[2]Note that we can leave out the question if $Q$ remains the same for all examples (e.g., Figure 2 left).

In this task, given a set of statements (verbalized rules and facts) and a question, the LM is asked to determine if the question statement is true or false. Annotations of reasoning trees $G$ are also provided in the dataset. Again, each tree has only one node at any height larger than 0. Thus, knowing the node height is sufficient to recover $G$. To simplify our analysis, we remove examples annotated with multiple reasoning trees. Details are in Appendix C.3. Furthermore, to avoid tree ambiguity (Appendix C.4), we only keep examples with reasoning trees of depth upto 1, which account for 70% of the data in ProofWriter.

**AI2 Reasoning Challenge (ARC).** The ARC dataset contains multiple-choice questions from middle-school science exams (Clark et al., 2018). However, the original dataset does not have reasoning tree annotations. Thus, we consider a subset of the dataset provided by Ribeiro et al. (2023), which annotates around 1000 examples. Considering the limited number of examples, we do not include analysis of finetuned LMs and mainly focus on the in-context learning setting. More details about the dataset are in Appendix C.3.

For both ProofWriter and ARC tasks, we select **LLaMA (7B)** (Touvron et al., 2023) as the LM for analysis. The tasks are formalized as classifications: predicting the answer token (e.g., *true* or *false* for ProofWriter).[3] We compare two settings: LLaMA with 4-shot in-context learning setting, and LLaMA finetuned with supervised signal (i.e., LLaMA_FT). We partially finetune LLaMA on attention parameters. Implementation details about in-context learning and finetuning on LLaMA can be found in Appendix C.1.

## 3 `MechanisticProbe`

In this section, we introduce `MechanisticProbe`: a probing approach that analyzes how LMs solve multi-step reasoning tasks by recovering reasoning trees from the attentions of LMs.

### 3.1 Problem Formulation

Our goal is to shed light on how LMs solve procedural reasoning tasks. Formally, given an LM with $L$ layers and $H$ attention heads, we assume that the LM can handle the multi-step reasoning task with sufficiently high accuracy.

---

[3]Note that most reasoning tasks with LMs can be formalized as the single-token prediction format (e.g., multi-choice question answering).

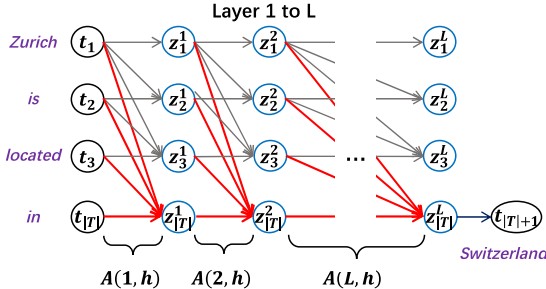

Figure 3: An illustration of the attention mechanism in a LM. The example sentence here is: *Zurich is located in Switzerland*. Token $t_{|T|}$ is the last token used for the prediction of the next token $t_{|T|+1}$. The arrows here show the attentions, where the red arrows denote the attentions to the last token.

Let us consider the simplest of the three tasks ($k$-th smallest element). Each statement $S_i$ here is encoded as one token $t_i$. The input text (containing all statements in $\mathcal{S}$) to the LM comprises a set of tokens as $T = (t_1, t_2, ...)$. We denote the hidden representations of the token $t_i$ at layer $l$ by $z_i^l$. We further denote the attention of head $h$ between $z_i^{l+1}$ and $z_j^l$ as $A(l, h)[i, j]$. As shown in Figure 3, the attention matrix at layer $l$ of head $h$ is denoted as $A(l, h)$, the lower triangular matrix.[4] The overall attention matrix then can be denoted by $A = \{A(l, h) | 1 \le l \le L; 1 \le h \le H\}$.

Our probing task is to detect $G$ from $A$, i.e. modeling $P(G|A)$. However, note that the size of $A$ is very large – $L \times H \times |T|^2$, and it could contain many redundant features. For example, if the number of tokens in the input is 100, the attention matrix $A$ for LLaMA contains millions of attention weights. It is impossible to probe the information directly from $A$. In addition, the tree prediction task is difficult (Hou and Sachan, 2021) and we want our probe to be simple (Belinkov, 2022) as we want it to provide reliable interpretation about the LM rather than learn the task itself. Therefore, we introduce two ways to simplify attentions and the probing task design.

### 3.2 Simplification of $A$

In general, we propose two ways to simplify $A$. Considering that LLaMA is a large LM, we propose extra two ways to further reduce the number of considered attention weights in $A$ for it.

**Focusing on the last token.** For causal LMs, the representation of the last token in the last layer $z_{|T|}^L$

---

[4]Note that in this work we only consider causal LMs.

is used to predict the next token $t_{|T|+1}$ (Figure 3). Thus, we simplify $\boldsymbol{A}$ by focusing on the attentions on the last input token, denoted as $\boldsymbol{A}_{\text{simp}}$. This reduces the size of attentions to $L \times H \times |T|$. Findings in previous works support that $\boldsymbol{A}_{\text{simp}}$ is sufficient to reveal the focus of the prediction token (Brunner et al., 2020; Geva et al., 2023). In our experimental setup, we also find that analysis on $\boldsymbol{A}_{\text{simp}}$ gives similar results to that on $\boldsymbol{A}$ (Appendix B.4).

**Attention head pooling.** Many existing attention-based analysis methods use pooling (e.g., mean pooling or max pooling) on attention heads for simplicity (Abnar and Zuidema, 2020; Manning et al., 2020; Murty et al., 2023). We follow this idea and take the mean value across all attention heads for our analysis. Then, the size of $\boldsymbol{A}_{\text{simp}}$ is further reduced to $L \times |T|$.

**Ignoring attention weights within the statement (LLaMA).** For LLaMA on the two natural language reasoning tasks, a statement $S_i$ could contain multiple tokens, i.e., $|T| >> |\mathcal{S}|$. Thus, the size of $\boldsymbol{A}_{\text{simp}}$ can still be large. To further simplify $\boldsymbol{A}_{\text{simp}}$ under this setting, we regard all tokens of a statement as a hypernode. That is, we ignore attentions within hypernodes and focus on attentions across hypernodes. As shown in Figure 4, we can get $\boldsymbol{A}_{\text{simp}}^{\text{cross}}$ via *mean pooling* on all tokens of a statement and *max pooling* on all tokens of $Q$ as:[5]

$$\boldsymbol{A}_{\text{simp}}^{\text{cross}}(l,h)[i] = \max_{t_{j'} \in Q} \left( \operatorname*{mean}_{t_{i'} \in S_i} \left( \boldsymbol{A}(l,h)[i',j'] \right) \right).$$

The size of simplified cross-hypernode attention matrix $\boldsymbol{A}_{\text{simp}}^{\text{cross}}$ is further reduced to $L \times (|\mathcal{S}| + 1)$.

**Pruning layers (LLaMA).** Large LMs (e.g., LLaMA) are very deep (i.e., $L$ is large), and are pretrained for a large number of tasks. Thus, they have many redundant parameters for performing the reasoning task. Inspired by Rogers et al. (2020), we prune the useless layers of LLaMA for the reasoning task and probe attentions of the remaining layers. Specifically, we keep a minimum number of layers that maintain the LM's performance on a held-out development set, and deploy our analysis on attentions of these layers. For 4-shot LLaMA, 13/15 (out of 32) layers are removed for

[5]We take mean pooling for statements since max pooling cannot differentiate statements with many overlapped words. We take max pooling for $Q$ since it is more differentiable for the long input text. In practice, users can select different pooling strategies based on their own requirements.

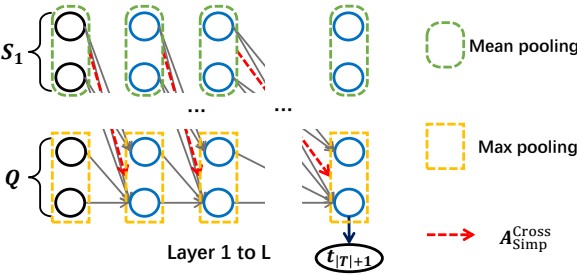

Figure 4: An illustration of the way to extend our attention simplification method. In this example, $S_1$ and $Q$ are both composed of 2 tokens. Thus, there are $2 \times 2$ attentions. We mean pool all the tokens within the statements (remaining $1 \times 2$ attentions), and max pool all the tokens in the question (remaining $1 \times 1$ attention).

ProofWriter and ARC respectively. For finetuned LLaMA, 18 (out of 32) layers are removed for ProofWriter. More details about the attention pruning can be found in Appendix C.2.

### 3.3 Simplification of the Probing Task

We simplify the problem of predicting $G$ by breaking it down into two classification problems: classifying if the statement is useful or not, and classifying the height of the statement in the tree.

$$P(G|\boldsymbol{A}_{\text{simp}}) = P(V|\boldsymbol{A}_{\text{simp}}) \cdot P(G|V, \boldsymbol{A}_{\text{simp}}).$$

Here, $V$ is the set of nodes in $G$. $P(V|\boldsymbol{A}_{\text{simp}})$ (binary classification) measures if LMs can select useful statements from the input based on attentions, revealing if LMs correctly focus on useful statements. Given the set of nodes in $G$, the second probing task is to decide the reasoning tree. We model $P(G|V, \boldsymbol{A}_{\text{simp}})$ as the multiclass classification for predicting the height of each node. For example, when the reasoning tree depth is 2, the height label set is $\{0, 1, 2\}$. Note that for the three reasoning tasks considered by us, $G$ is always a simple tree that has multiple leaf nodes but one intermediate node at each height.[6]

### 3.4 Probing Score

To limit the amount of information the probe learns about the probing task, we use a non-parametric classifier: k-nearest neighbors (kNN) to perform the two classification tasks. We use $S_{\text{F1}}(V|\boldsymbol{A}_{\text{simp}})$ and $S_{\text{F1}}(G|V, \boldsymbol{A}_{\text{simp}})$ to denote their F1-Macro scores. To better interpret the probing results, we

[6]In practice, many reasoning graphs can be formalized as the chain-like tree structure (Wei et al., 2022). We leave the analysis with more complex $G$ for future work.

introduce a random probe baseline as the control task (Hewitt and Liang, 2019) and instead of looking at absolute probing values, we interpret the probing scores compared to the score of the random baseline. The probe scores are defined as:

$$S_{\text{P1}} = \frac{S_{\text{F1}}(V|\boldsymbol{A}_{\text{simp}}) - S_{\text{F1}}(V|\boldsymbol{A}_{\text{rand}})}{1 - S_{\text{F1}}(V|\boldsymbol{A}_{\text{rand}})}, \quad (1)$$

$$S_{\text{P2}} = \frac{S_{\text{F1}}(G|V, \boldsymbol{A}_{\text{simp}}) - S_{\text{F1}}(G|V, \boldsymbol{A}_{\text{rand}})}{1 - S_{\text{F1}}(G|V, \boldsymbol{A}_{\text{rand}})}, \quad (2)$$

where $\boldsymbol{A}_{\text{rand}}$ is the simplified attention matrix given by a randomly initialized LM. After normalization, we have the range of our probing scores: $S_{\text{P1}}, S_{\text{P2}} \in [0, 1]$. Small values mean that there is no useful information about $G$ in attention, and large values mean that the attention patterns indeed contain much information about $G$.

## 4  Mechanistic Probing of LMs

We use the probe to analyze how LMs perform the reasoning tasks. We first verify the usefulness of attentions in understanding the reasoning process of LMs by visualizing $\boldsymbol{A}_{\text{simp}}$ (§4.1). Then, we use the probe to quantify the information of $G$ contained in $\boldsymbol{A}_{\text{simp}}$ (§4.2). Finally, we report layer-wise probing results to understand if LMs are reasoning procedurally across their architecture (§4.3). [7]

### 4.1  Attention Visualization

We first analyze $\boldsymbol{A}_{\text{simp}}$ on the $k$-th smallest element task via visualizations. We permute $\boldsymbol{A}_{\text{simp}}$ arranging the numbers in ascending order and denote this permulation as $\pi(\boldsymbol{A}_{\text{simp}})$. We show visualizations of $\mathbb{E}[\pi(\boldsymbol{A}_{\text{simp}})]$ on the test data in Figure 5. We observe that when GPT-2 tries to find the $k$-th smallest number, the prediction token first focuses on top-$k$ numbers in the list with bottom layers. Then, the correct answer is found in the top layers. These findings suggest that GPT-2 solves the reasoning task in two steps following the reasoning tree $G$. We further provide empirical evidence in Appendix B.4 to show that analysis on $\boldsymbol{A}_{\text{simp}}$ gives similar conclusions compared to that on $\boldsymbol{A}$.

### 4.2  Probing Scores

Next, we use our MechanisticProbe to quantify the information of $G$ contained in $\boldsymbol{A}_{\text{simp}}$ (GPT-2)

---

or $\boldsymbol{A}_{\text{simp}}^{\text{cross}}$ (LLaMA).

**GPT-2 on $k$-th smallest element ($\boldsymbol{A}_{\text{simp}}$).** We consider two versions of GPT-2: a pretrained version and a finetuned version (GPT-2$_{\text{FT}}$). We report our two probing scores (Eq. 1 and Eq. 2) with different $k$ in Table 1. The unnormalized F1-macro scores (i.e., $S_{\text{F1}}(V|\boldsymbol{A}_{\text{simp}})$ and $S_{\text{F1}}(G|V, \boldsymbol{A}_{\text{simp}})$) can be found in Appendix B.2.

Table 1: Probing scores for GPT-2 models on synthetic reasoning tasks with different $k$. We also provide the test accuracy of these finetuned models for reference. Note that when $k = 1$, the depth of $G$ is 0, and $S_{\text{F1}}(G|V, \boldsymbol{A}_{\text{simp}})$ is always equal to 1. Thus, we leave these results blank. Results show that we can clearly detect $G$ from attentions of GPT-2$_{\text{FT}}$.

| $k$ | Test Acc. | | $S_{\text{P1}}$ | | $S_{\text{P2}}$ | |
|---|---|---|---|---|---|---|
| | GPT-2 | GPT-2$_{\text{FT}}$ | GPT-2 | GPT-2$_{\text{FT}}$ | GPT-2 | GPT-2$_{\text{FT}}$ |
| 1 | | 99.63 | 7.09 | 92.94 | - | - |
| 2 | | 99.42 | 5.88 | 93.71 | 20.65 | 98.05 |
| 3 | | 98.23 | 13.48 | 91.62 | 13.59 | 95.76 |
| 4 | 0.00 | 94.89 | 20.73 | 88.34 | 8.04 | 92.52 |
| 5 | | 93.38 | 24.15 | 87.54 | 13.81 | 88.17 |
| 6 | | 92.62 | 24.82 | 86.89 | 11.18 | 95.06 |
| 7 | | 91.37 | 25.87 | 76.61 | < 1 | 91.56 |
| 8 | | 91.29 | 22.30 | 78.73 | 1.06 | 93.93 |

These results show that without finetuning, GPT-2 is incapable of solving the reasoning task, and we can detect little information about $G$ from GPT-2's attentions.[8] However, for GPT-2$_{\text{FT}}$, which has high test accuracy on the reasoning task, MechanisticProbe can easily recover the reasoning tree $G$ from $\boldsymbol{A}_{\text{simp}}$. This further confirms that GPT-2$_{\text{FT}}$ solves this synthetic reasoning task following $G$ in Figure 2 (left).

**LLaMA on ProofWriter and ARC ($\boldsymbol{A}_{\text{simp}}^{\text{cross}}$).** Similarly, we use MechanisticProbe to probe LLaMA on the two natural language reasoning tasks. For efficiency, we randomly sampled 1024 examples from the test sets for our analysis. When depth = 0, LLaMA only needs to find out the useful statements for reasoning ($S_{\text{P1}}$). When depth=1, LLaMA needs to determine the next reasoning step.

Probing results with different numbers of input statements (i.e., $|\mathcal{S}|$) are in Table 2. Unnormalized classification scores can be found in Appendix C.5. It can be observed that all the probing scores are much larger than 0, meaning that the attentions indeed contain information about $G$.

---

[7] We have additional empirical explorations in the Appendix. Appendix B.6 shows that different finetuning methods would not influence probing results. Appendix B.7 explores how the reasoning task difficulty and LM capacity influence the LM performance. Appendix B.8 researches the relationships between GPT-2 performance and our two probing scores.

[8] Visualization of $\mathbb{E}[\pi(\boldsymbol{A}_{\text{simp}})]$ for (pretrained) GPT-2 can be found in Appendix B.3. It shows that GPT-2 can somehow find out the largest number from the number list.

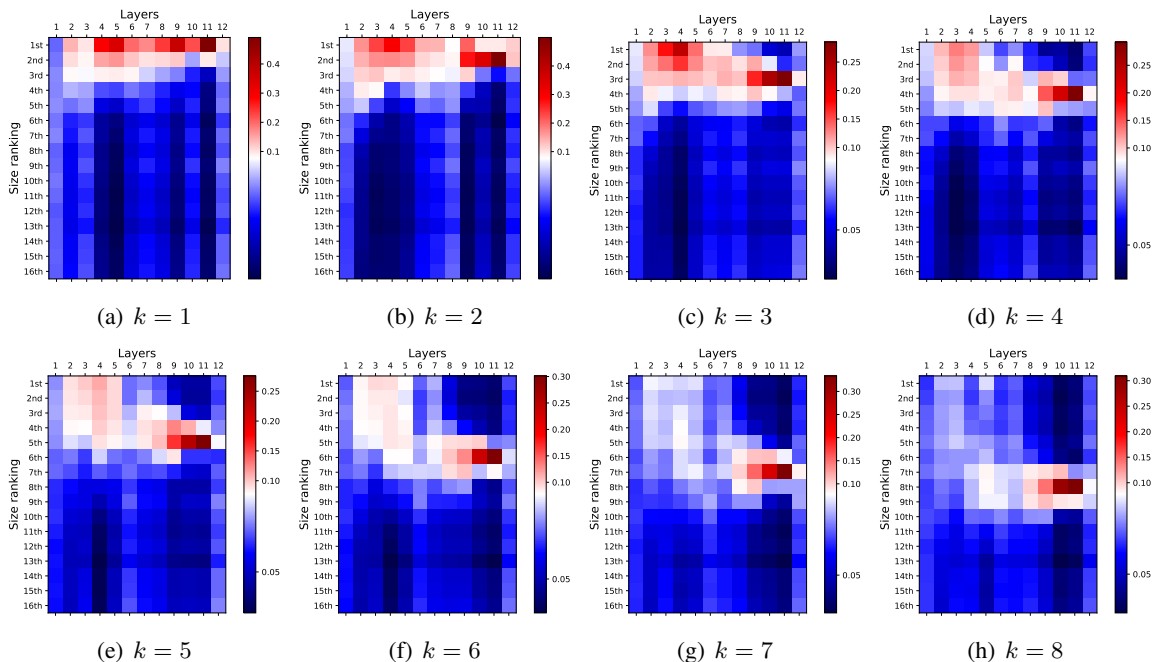

Figure 5: Visualization of $\mathbb{E}[\pi(\boldsymbol{A}_{\text{simp}})]$ for $k$ ranging from 1 to 8. Note that $m = 16$ here, and when $k > 8$, the reasoning task becomes finding $k$-th largest number. The visualizations are roughly the same without loss of generality. The $x$-axis represents the layer index and the $y$-axis represents the size (rank) of the number. The value of each cube is the attention weight (better view in color). Results show that layers of finetuned GPT-2 have different functions: bottom layers focusing on top-$k$ numbers and top layers focusing on $k$-smallest numbers.

Table 2: Probing results for LLaMA on natural language reasoning. Note that when depth $= 0$, $S_{\text{F1}}(G|V, \boldsymbol{A}_{\text{simp}})$ is always equal to 1. When depth $= 1$ and $|\mathcal{S}| = 2$, all input statements are useful. $S_{\text{F1}}(V|\boldsymbol{A}_{\text{simp}})$ is always equal to 1. We leave these results blank. Regarding ARC, we only report the scores under the 4-shot setting. Results show that attentions of LLaMA contain some information about selecting useful statements for ProofWriter but much information for ARC. Regarding determining reasoning steps, attentions of LLaMA contain much information for both ProofWriter and ARC.

| | | ProofWriter | | | | | |
|---|---|---|---|---|---|---|---|
| Depth | $|\mathcal{S}|$ | Test Acc. | | $S_{\text{P1}}$ | | $S_{\text{P2}}$ | |
| | | LLaMA | LLaMA$_{\text{FT}}$ | LLaMA | LLaMA$_{\text{FT}}$ | LLaMA | LLaMA$_{\text{FT}}$ |
| 0 | All | 81.72 | | 57.21 | 49.08 | - | - |
| | 2 | 94.81 | | - | - | 100 | 100 |
| | 4 | 95.12 | | 44.83 | 48.14 | 93.34 | 96.22 |
| | 8 | 92.19 | 100 | 27.39 | 40.09 | 83.75 | 96.44 |
| 1 | 12 | 90.53 | | 26.23 | 32.70 | 77.58 | 93.45 |
| | 16 | 89.55 | | 17.18 | 21.07 | 77.85 | 89.31 |
| | 20 | 88.38 | | 11.10 | 15.84 | 79.99 | 94.11 |
| | 24 | 86.13 | | 9.39 | 17.33 | 80.32 | 94.42 |
| | | ARC | | | | | |
| 1 | - | 56.32 | | 97.49 | - | 61.73 | - |
| 2 | - | 55.41 | - | 96.49 | - | 53.40 | - |

Looking at $S_{\text{P1}}$ on ProofWriter, when the number of input statements is small, MechanisticProbe can clearly decide the useful statements based on attentions. However, it becomes harder when there are more useless statements (i.e., $|S|$ is large). However, for ARC, our probe can always detect useful

statements from attentions easily.

Looking at $S_{\text{P2}}$, we notice that our probe can easily determine the height of useful statements based on attentions on both ProofWriter and ARC datasets. By comparing the probing scores on ProofWriter, we find that LLaMA$_{\text{FT}}$ always has higher probing scores than 4-shot LLaMA, implying that finetuning with supervised signals makes the LM to follow the reasoning tree $G$ more clearly. We also notice that 4-shot LLaMA is affected more by the number of useless statements than LLaMA$_{\text{FT}}$, indicating a lack of robustness of reasoning in the few-shot setting.

### 4.3 Layer-wise Probing

After showing that LMs perform reasoning following oracle reasoning trees, we investigate how this reasoning happens inside the LM layer-by-layer.

**GPT-2 on $k$-th smallest element.** In order to use our probe layer-by-layer, we define the set of simplified attentions before layer $l$ as $\boldsymbol{A}_{\text{simp}}(: l) = \{\boldsymbol{A}_{\text{simp}}(l')|l' \leq l\}$. Then, we report our probing scores $S_{\text{P1}}(l)$ and $S_{\text{P2}}(l)$ on these partial attentions from layer 1 to layer 12. We denote these layer-wise probing scores as $S_{\text{F1}}(V|\boldsymbol{A}_{\text{simp}}(: l))$ and $S_{\text{F1}}(G|V, \boldsymbol{A}_{\text{simp}}(: l))$.

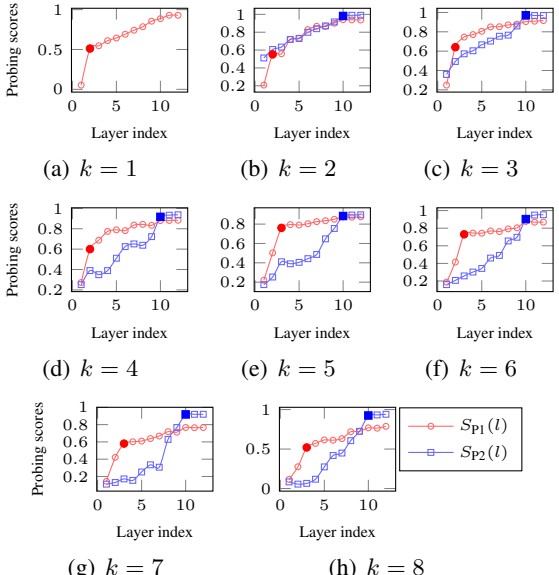

(a) $k = 1$    (b) $k = 2$    (c) $k = 3$

(d) $k = 4$    (e) $k = 5$    (f) $k = 6$

(g) $k = 7$    (h) $k = 8$

Figure 6: Two probing scores on partial attentions across 12 layers. The $x$-axis represents the layer index of GPT-$2_{\text{FT}}$, and the $y$-axis represents the probing scores. Similarly, we test different $k$ from 1 to 8 ($m = 16$ by default). Results show that each layer of GPT-$2_{\text{FT}}$ focuses on different steps of the synthetic reasoning task.

Figure 6 shows the layer-wise probing scores for each $k$ for GPT-$2_{\text{FT}}$ models. Observing $S_{\text{P1}}$, i.e., selecting top-$k$ numbers, we notice that GPT-$2_{\text{FT}}$ quickly achieves high scores in initial layers and then, $S_{\text{P1}}$ increases gradually. Observing $S_{\text{P2}}$, i.e., selecting the $k$-th smallest number from top-$k$ numbers, we notice that GPT-$2_{\text{FT}}$ does not achieve high scores until layer 10. This reveals how GPT-$2_{\text{FT}}$ solves the task internally. The bottom layers find out the top-$k$ numbers, and the top layers select the $k$-th smallest number among them. Results in Figure 5 also support the above findings.

**LLaMA on ProofWriter and ARC ($A_{\text{simp}}^{\text{cross}}$).** Similarly, we report layer-wise probing scores $S_{\text{P1}}(l)$ and $S_{\text{P2}}(l)$ for LLaMA under the 4-shot setting. We further report $S_{\text{F1}}(V_{\text{height}}|A_{\text{simp}}^{\text{cross}}(:l))$ for nodes with height $= 0$ and height $= 1$ to show if the statement at height 0 is processed in LLaMA before the statement at height 1.

The layer-wise probing results for ProofWriter are shown in Figure 7(a-e). We find that similar to GPT-2, probing results for 4-shot LLaMA reach a plateau at an early layer (layer 2 for $S_{\text{P1}}(l)$), and at middle layers for $S_{\text{P2}}(l)$. This obserrvation holds as we vary $|\mathcal{S}|$ from 4 to 20. This shows that similar to GPT-2, LLaMA first tries to identify useful statements in the bottom layers.

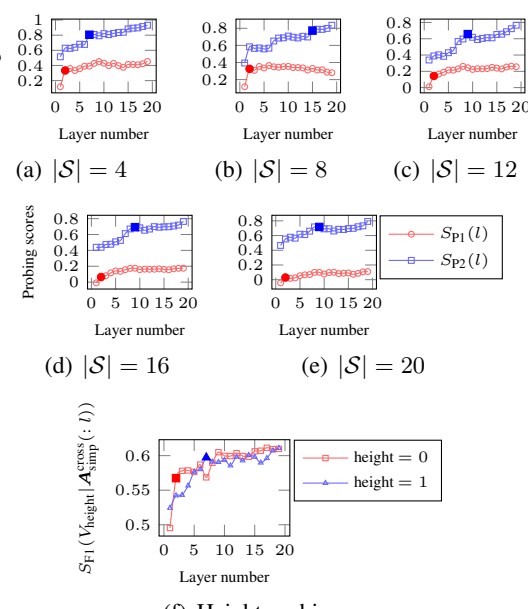

(a) $|\mathcal{S}| = 4$    (b) $|\mathcal{S}| = 8$    (c) $|\mathcal{S}| = 12$

(d) $|\mathcal{S}| = 16$    (e) $|\mathcal{S}| = 20$

(f) Height probing

Figure 7: Layer-wise probing results on ProofWriter. $x$-axis represents the number of layers used for probing. We can find that $S_{\text{P1}}(l)$ scores reach the plateau quickly and $S_{\text{P2}}(l)$ scores increase smoothly till the middle layers. This indicates that useful statement selection is mainly finished in the bottom layers and the reasoning step is decided in the middle layers. Results of $S_{\text{F1}}(V_{\text{height}}|A_{\text{simp}}^{\text{cross}}(:l))$ shows that statements at height 0 are identified by LLaMA in bottom layers, and statement at height 1 are identified later in middle layers.

Then, it focuses on predicting the reasoning steps given the useful statements. Figure 7(f) shows how $S_{\text{F1}}(V_{\text{height}}|A_{\text{simp}}^{\text{cross}}(:l))$ varies across layers. LLaMA identifies the useful statements from all the input statements (height 0) immediately in layer 2. Then, LLaMA gradually focuses on these statements and builds the next layer of the reasoning tree (height 1) in the middle layers.

The layer-wise probing results for ARC are in Figure 8. Similar to ProofWriter, the $S_{\text{P1}}(l)$ scores on ARC for 4-shot LLaMA reach a plateau at an early layer (layer 2), and at middle layers for $S_{\text{P2}}(l)$. This also shows that LLaMA tries to identify useful statements in the bottom layers and focuses on the next reasoning steps in the higher layers.

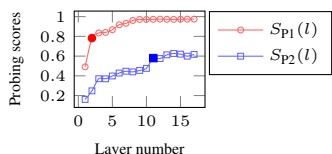

Figure 8: Layer-wise probing results on ARC (depth=1). Similar to that of ProofWriter, we can find that $S_{\text{P1}}(l)$ scores reach the plateau quickly and $S_{\text{P2}}(l)$ scores increase smoothly till the middle layers.

## 5 Do LMs Reason Using $A_{\text{simp}}$?

Our analysis so far shows that LMs encode the reasoning trees in their attentions. However, as argued by Ravichander et al. (2021); Lasri et al. (2022); Elazar et al. (2021), this information might be accidentally encoded but not actually used by the LM for inference. Thus, we design a causal analysis for GPT-2 on the $k$-th smallest element task to show that LMs indeed perform reasoning following the reasoning tree. The key idea is to prove that the attention heads that contribute to $A_{\text{simp}}$ are useful to solve the reasoning task, while those heads that are irrelevant (i.e., independent) to $A_{\text{simp}}$ are not useful in the reasoning task.

Intuitively, for the $k$-th smallest element task, attention heads that are sensitive to the number size (rank) are useful, while heads that are sensitive to the input position are not useful. Therefore, for each head, we calculate the attention distribution on the test data to see if the head specially focuses on numbers with a particular size or position. We use the entropy of the attention distribution to measure this[9]: small entropy means that the head focuses particularly on some numbers. We call the entropy with respect to number size as *size entropy* and that with respect to input position as *position entropy*. The entropy of all the heads in terms of number size and position can be found in Appendix B.5.

We prune different kinds of attention heads in order of their corresponding entropy values and report the test accuracy on the pruned GPT-2.[10] Results with different $k$ are shown in Figure 9. We find that the head with a small size entropy is essential for solving the reasoning task. Dropping $10\%$ of this kind of head, leads to a significant drop in performance on the reasoning task. The heads with small position entropy are highly redundant. Dropping $40\%$ of the heads with small position entropy does not affect the test accuracy much. Especially when $k = 1$, dropping $90\%$ position heads could still promise a high test accuracy.

These results show that heads with small size entropy are fairly important for GPT-2 to find $k$-th smallest number while those with small position entropy are useless for solving the task. Note that the reasoning tree $G$ is defined on the input number

---

[9]We regard the attentions $\mathbb{E}[A_{\text{simp}}(l, h)]$ as probabilities, and calculate the entropy of head $h$ at layer $l$ using them.

[10]We gradually prune heads, and the order of pruning heads is based on the size/position entropy. For example, for the *size entropy* pruning with $50\%$ pruned heads, we remove attention heads that have the top $50\%$ smallest size entropy.

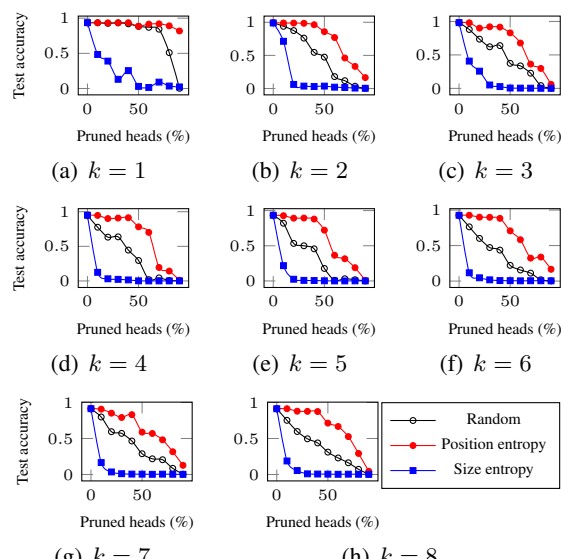

(a) $k = 1$    (b) $k = 2$    (c) $k = 3$
(d) $k = 4$    (e) $k = 5$    (f) $k = 6$
(g) $k = 7$       (h) $k = 8$

Figure 9: Test accuracy on the reasoning task for different pruning rates. The $x$-axis represents pruning rates: ranging from $0\%$ to $90\%$, and the $y$-axis represents the test accuracy. We prune heads in the ascending order of their size/position entropy. Results show that heads with small size entropy are essential to the test accuracy while those with small position entropy are useless.

size and it is independent of the number position. MechanisticProbe detects the information of $G$ from attentions. Thus, our probing scores would be affected by the heads with small size entropy but would not be affected by heads with small position entropy. Then, we can say that changing our probing scores (via pruning heads in terms of size entropy) would cause the test accuracy change. Therefore, we say that there is a causal relationship between our probing scores and LM performance, and LMs perform reasoning following the reasoning tree in $A_{\text{simp}}$.

## 6 Correlating Probe Scores with Model Accuracy and Robustness

Our results show that LMs indeed reason mechanistically. But, is mechanistic reasoning necessary for LM performance or robustness? We attempt to answer this question by associating the probing scores with the performance and robustness of LMs. Given that finetuned GPT-2 has a very high test accuracy on the synthetic task and LLaMA does not perform as well on ARC, we conduct our analysis mainly with LLaMA on the ProofWriter task.

**Accuracy.** We randomly sample 64 to 128 examples from the dataset and test 4-shot LLaMA on these examples. We calculate their test accuracy

Table 3: Pearson correlation coefficient between our probing scores and test accuracy for 4-shot LLaMA.

| Pearson correlation coefficient $\rho$ ($\times 100\%$) | |
| --- | --- |
| $\rho\,(S_{P1}, S_{P2})$ | 0.01 |
| $\rho\,(\text{Test accuracy}, S_{P1})$ | 27.42 |
| $\rho\,(\text{Test accuracy}, S_{P2})$ | 71.13 |

and the two probing scores $S_{P1}$ and $S_{P2}$. We repeat this experiment $2048$ times. Then, we calculate the correlation between the probing scores and test accuracy. From Table 3, we find that test accuracy is closely correlated with $S_{P2}$. This implies that when we can successfully detect reasoning steps of useful statements from LM's attentions, the model is more likely to produce a correct prediction.

**Robustness.** Following the same setting, we also associate the probing scores with LM robustness. In order to quantify model robustness, we randomly corrupt one useless input statement for each example, such that the prediction would remain unchanged.[11] We measure robustness by the decrease in test accuracy after the corruption.

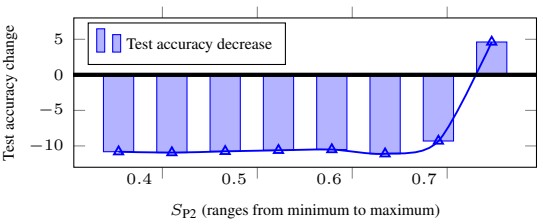

Figure 10: Histogram for LM prediction robustness and $S_{P2}$. We measure robustness by the change in accuracy on the corrupted dataset. We present results with $8$ bins. Results show that LLaMA is more robust to noise on examples with higher probing scores $S_{P2}$.

Figure 10 shows that if $S_{P2}$ is small (less than 0.7), the prediction of LLaMA could be easily influenced by the noise (test accuracy decreases around $10\%$). However, if the probing $S_{P2}$ is high, LLaMA is more robust, i.e., more confident in its correct prediction (test accuracy increases around $4\%$). This provides evidence that if the LM encodes the gold reasoning trees, its predictions are more reliable (i.e., robust to noise in the input).

## 7  Related Work

**Attention-based analysis of LMs.** Attention has been popularly used to interpret LMs (Vig and Belinkov, 2019; DeRose et al., 2021; Bibal et al., 2022). A direct way for interpreting an attention-based model is to visualize attentions (Samaran et al., 2021; Chefer et al., 2021). But the irrelevant, redundant, and noisy information captured by attentions makes it hard to find meaningful patterns. Alternatively, accumulation attentions that quantify how information flows across tokens can be used for interpretation (Abnar and Zuidema, 2020; Eberle et al., 2022). However, for casual LMs, information flows in one direction, and it causes an over-smoothing problem when the model is deep.[12] To tackle this, other works propose new metrics and analyze attentions using them (Ethayarajh and Jurafsky, 2021; Liu et al., 2022). However, these metrics are proposed under specific scenarios, and these are not useful for detecting the reasoning process in LMs. We address this challenge by designing a more structured probe that predicts the reasoning tree in LMs.

**Mechanistic Interpretability.** Mechanistic interpretation explains how LMs work by reverse engineering, i.e., reconstructing LMs with different components (Räuker et al., 2022). A recent line of work provides interpretation focusing on the LM's weights and intermediate representations (Olah et al., 2017, 2018, 2020). Another line of work interprets LMs focusing on how the information is processed inside LMs (Olsson et al., 2022; Nanda et al., 2023). Inspired by them, Qiu et al. (2023) attempts to interpret how LMs perform reasoning. However, existing explorations do not cover the research problem we discussed.

## 8  Conclusion

In this work, we raised the question of whether LMs solve procedural reasoning tasks step-by-step within their architecture. In order to answer this question, we designed a new probe that detects the oracle reasoning tree encoded in the LM architecture. We used the probe to analyze GPT-2 on a synthetic reasoning task and the LLaMA model on two natural language reasoning tasks. Our empirical results show that we can often detect the information in the reasoning tree from the LM's attention patterns, lending support to the claim that LMs may indeed be reasoning "mechanistically".

---

[11]We do it by adding negation: corrupting $S_i$ as ["That"$\circ$ $S_i \circ$ " is false"], where $\circ$ means string concatenation.

[12]The issue is that all token representations are dominated by the first token. Detailed discussion is in Appendix A.

## Limitations

One key limitation of this work is that we considered fairly simple reasoning tasks. We invite future work to understand the mechanism behind LM-based reasoning by exploring more challenging tasks. We list few other limitations of our work below:

**Mutli-head attention.** In this work, most of our analysis takes the mean value of attentions across all heads. However, we should notice that attention heads could have different functions, especially when the LM is shallow but wide (e.g., with many attention heads, and very high-dimensional hidden states). Shallow models might still be able to solve procedural reasoning tasks within a few layers, but the functions of the head could not be ignored.

**Auto-regressive reasoning tasks.** In our analysis, we formalize the reasoning task as classification, i.e., single-token prediction. Thus, the analysis could only be deployed on selected reasoning tasks. Some recent reasoning tasks are difficult and can only be solved by LMs via chain-of-thought prompting. We leave the analysis of reasoning under this setting for future work.

## Acknowledgements

We are grateful to anonymous reviewers for their insightful comments and suggestions. Yifan Hou is supported by the Swiss Data Science Center PhD Grant (P22-05) and Alessandro Stolfo is supported by armasuisse Science and Technology through a CYD Doctoral Fellowship. Antoine Bosselut gratefully acknowledges the support of the Swiss National Science Foundation (No. 215390), Innosuisse (PFFS-21-29), the EPFL Science Seed Fund, the EPFL Center for Imaging, Sony Group Corporation, and the Allen Institute for AI. Mrinmaya Sachan acknowledges support from the Swiss National Science Foundation (Project No. 197155), a Responsible AI grant by the Haslerstiftung; and an ETH Grant (ETH-19 21-1).

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

## A  Proof of The First-Token Domination

*Proof.* Without loss of generality, we assume the LM have $L$ layers with only 1 attention head ($h = H = 1$), and the attention weight matrix in layer $l$ is $\boldsymbol{A}(l, h)$. We consider the input that have more than 1 token. We model attention as flows following the setting of Abnar and Zuidema (2020). Then, the attention accumulation Accum() of the token $\boldsymbol{z}_i^{l+1}$ in the layer $l + 1$ can be written as

$$\text{Accum}(\boldsymbol{z}_i^{l+1}) = \sum_{1 \leq j \leq |T|} \boldsymbol{a}_{i,j}(l, h) \cdot \text{Accum}(\boldsymbol{z}_j^l), \tag{3}$$

where we have

$$\text{Accum}(\boldsymbol{z}_i^1) = \sum_{1 \leq j \leq |T|} \boldsymbol{a}_{i,j}(l, h) \cdot t_j. \tag{4}$$

Since $\boldsymbol{a}_{i,j}(l, h)$ is the attention weight for casual LM, we have

$$\begin{cases} \boldsymbol{a}_{i,j}(l, h) = 0 & \text{if} \quad i < j \\ 0 < \boldsymbol{a}_{i,j}(l, h) < 1 & \text{if} \quad i \geq j \\ \sum_j \boldsymbol{a}_{i,j}(l, h) = 1 \end{cases} \tag{5}$$

Note that attention is normalized by *Softmax* function in LMs. The minimum attention weight is non-zero, and we assume there exist a constant $\epsilon > 0$ such as $\epsilon \leq \boldsymbol{a}_{i,j}(l, h) < 1$ if $i \geq j$. Now we define the information ratio $\text{IR}_{\boldsymbol{z}_i^l}(t_j)$ as the information of token $t_j$ stored in the hidden representation $\boldsymbol{z}_i^l$. Consider that in each layer, token $t_1$ would propagate its information to all tokens in the next layer with at least $\epsilon$ amount. Then, by tracing the information flow from other tokens $j > 1$, we have

$$1 - \text{IR}_{\boldsymbol{z}_i^{l+1}}(t_1) \leq (1 - \epsilon)(1 - \text{IR}_{\boldsymbol{z}_i^l}(t_1)). \tag{6}$$

Using the chain rule, we have

$$1 - \text{IR}_{\boldsymbol{z}_i^L}(t_1) \leq (1 - \epsilon)^L (1 - \text{IR}_{\boldsymbol{z}_i^1}(t_1)), \tag{7}$$

which means

$$\text{IR}_{\boldsymbol{z}_i^L}(t_1) \geq 1 - (1 - \epsilon)^{L-1}(1 - \text{IR}_{\boldsymbol{z}_i^1}(t_1)), \tag{8}$$

$$\text{IR}_{\boldsymbol{z}_i^L}(t_1) \geq 1 - (1 - \epsilon)^L. \tag{9}$$

With the inequality above, we know that if the LM is deep, i.e., $L$ is large, we have $\text{IR}_{\boldsymbol{z}_i^L}(t_1)$ increase exponentially in terms of layer $L$, which means that $\text{IR}_{\boldsymbol{z}_i^L}(t_1) \approx 1$ with large $L$ in general. $\square$

## B  Supplementary about GPT-2

We provide some more details on our experiments on GPT-2 as well as LLaMA to help in reproducibility. First of all, we fix the random seed (42) and use the same random seed for all experiments, including LM finetuning and interpretations. In addition, to make sure the random seed is unbiased, we further re-run the same experiment with different random seeds. All of our experiments have roughly the same results as those of using other seeds.

Second, we design our analysis as simply as possible to ensure that there is as little random influence (i.e., confounder) as possible. For MechanisticProbe, we select the kNN classifier. For LLaMA, we run analysis experiments on a 4-shot in-context learning setting.

Third, we report as many intermediate and supplement results as possible. In the Appendix, there are many other interesting findings. However, due to the space limit, we cannot present them in the main paper. We hope our findings are helpful to the community to better understand LMs.

### B.1  GPT-2 Finetuning

The finetuning settings for all GPT-2 models are roughly identical. We generate $0.98$ million sequences of numbers as the training data and $10,000$ in data for validation and testing. Note that the collision probability is extremely small, thus we can assume that there is no data leakage. The epoch number is set as 2, and the batch size here is $256$. We use the AdamW (Loshchilov and Hutter, 2019) optimizer with weight decay $1e - 3$ from Huggingface[13] for finetuning. The learning rate is $1e - 6$.

### B.2  Original Probing Scores

The original classification scores (F1-Macro) for two probing tasks can be found in Table 4. Here, *random* means we randomly initialize GPT-2 and use its attentions for probing as the random baseline. The other two *pretrained* and *finetuned* are the pretrained GPT-2 model and GPT-2 model after finetuning with supervised signals.

### B.3  Visualization of $\mathbb{E}[\pi(\boldsymbol{A}_{\text{simp}})]$ for GPT-2

We visualize the $\boldsymbol{A}_{\text{simp}}$ for pretrained GPT-2 without finetuning in Figure 11. We can find that even if pretrained GPT-2 cannot solve the synthetic reasoning task (finding $k$-th smallest number from a list). It can still somehow differentiate the size of

---

[13]https://huggingface.co/

Table 4: The original classification F1-Macro scores of GPT-2 for two probing tasks.

| Probing task | $S_{\mathrm{F1}}(V|\boldsymbol{A}_{\mathrm{simp}})$ | | | $S_{\mathrm{F1}}(G|V,\boldsymbol{A}_{\mathrm{simp}})$ | | |
|---|---|---|---|---|---|---|
| GPT-2 | random | pretrained | finetuned | random | pretrained | finetuned |
| $k=1$ | 48.38 | 52.04 | 96.36 | 100 | 100 | 100 |
| $k=2$ | 48.60 | 51.61 | 96.77 | 73.18 | 78.72 | 99.47 |
| $k=3$ | 47.62 | 54.68 | 95.61 | 61.45 | 66.69 | 98.36 |
| $k=4$ | 47.42 | 58.32 | 93.87 | 55.71 | 59.28 | 96.69 |
| $k=5$ | 48.50 | 60.94 | 93.58 | 54.86 | 55.48 | 94.66 |
| $k=6$ | 48.66 | 61.40 | 93.27 | 50.44 | 55.98 | 97.55 |
| $k=7$ | 49.28 | 62.40 | 88.14 | 51.75 | 51.11 | 97.55 |
| $k=8$ | 49.74 | 60.95 | 89.31 | 50.54 | 51.07 | 97.00 |

numbers. The largest number of the list often has slightly larger attentions in layer 11.

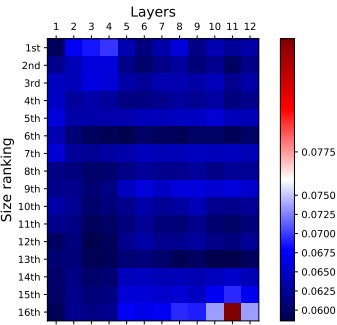

Figure 11: Visualization of $\mathbb{E}[\pi(\boldsymbol{A}_{\mathrm{simp}})]$ for pretrained GPT-2 (w/o any finetuning). Similarly, we take mean pooling for different attention heads.

### B.4 Visulization of $A$

To avoid disturbance, we remove $40\%$ position heads (heads with small position entropy) and take mean pooling on all left heads. We visualize $\boldsymbol{A}$ to directly show that $\boldsymbol{A}_{\mathrm{simp}}$ contains sufficient essential information of $\boldsymbol{A}$. We consider a special case when $k=2$, the largest number is at position 8 and the second largest number is at position 12.[14]

The attention $\mathbb{E}[\boldsymbol{A}]$ is visualized as in Figure 12. From Figure 12, we can get similar conclusion as on the visualization of $\mathbb{E}[\boldsymbol{A}_{\mathrm{simp}}]$. In the bottom layers, most hidden representations focus on top-2 numbers. In the top layers, most hidden representations focus on the second smallest number. This result proves that our analysis on $\boldsymbol{A}_{\mathrm{simp}}$ is as reasonable as that on $\boldsymbol{A}$.

We also provide the visualization of $\mathbb{E}[\boldsymbol{A}]$ on normal test data (i.e., the input position is independent of the size) in Figure 13. We can find that the attention distribution is even, and there is no

---

[14]We modify our test data only to satisfy this condition.

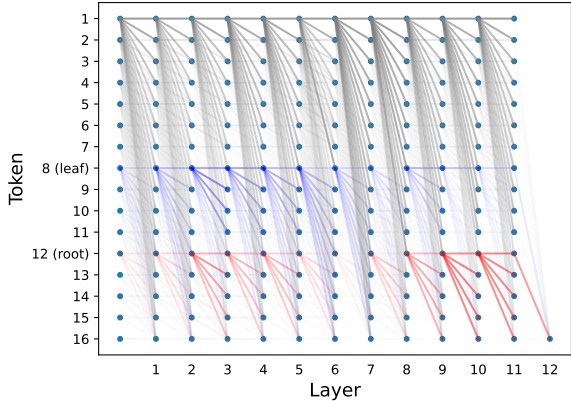

Figure 12: Visualization of $\mathbb{E}[\boldsymbol{A}]$. The $x$-axis represents layers and the $y$-axis represents input token positions. We take the mean pooling of $60\%$ attention heads with large position entropy. The leaf node (i.e., the largest number) is always at position 8, and the root node (i.e., the second largest number) is always at position 12. We track their attentions and visualize them by blue and red lines.

typical tendency.

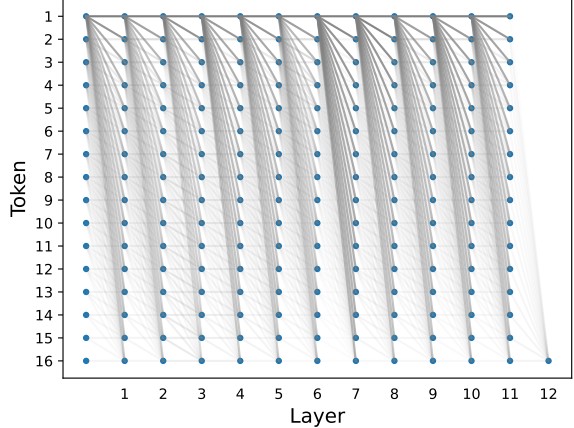

Figure 13: Visualization of $\mathbb{E}[\boldsymbol{A}]$ on normal test data. The $x$-axis represents layers and the $y$-axis represents input token positions. We take mean pooling of $60\%$ attention heads with large position entropy.

### B.5 Attention Head Entropy

From Figure 14, we can find that most heads belong to either position head or size head. Note that we generate input data randomly, thus, the number size and input position are independent of each other. Thus, one head cannot be both position head and size head.

### B.6 Do Finetuning Methods Matter?

To show that our analysis is robust, we explore the attention of GPT-2 with different ways of param-

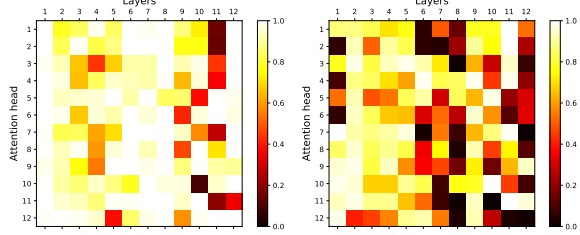

(a) Entropy in terms of num-ber size (rank)  (b) Entropy in terms of number input position

Figure 14: Entropy for all attention heads of finetuned GPT-2. $x$-axis represents layers and $y$-axis represents the head index. The cube (better view in color) shows the entropy value.

eter efficient finetuning (Ruder et al., 2022). We report our two probing scores $S_{P1}$ and $S_{P2}$ with various ways of finetuning in Figure 15 (We consider the condition when $k = 2$ and $m = 16$). We find that probing scores of finetuning the full GPT-2 model are similar to that of partially finetuning on attention parameters and MLP (multilayer percep-tron) parameters. These consistent results ensure the general usage of our MechanisticProbe on current large LMs with partial finetuning.

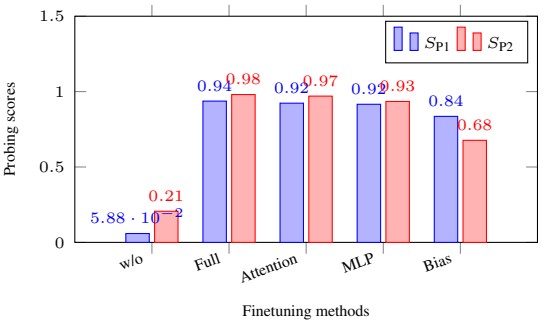

Figure 15: Probing scores of GPT-2 with different fine-tuning methods. Here, *w/o* denotes the baseline when GPT-2 model is not finetuned. *Full* denotes finetuning with all parameters. Other models are partial finetuned with corresponding parameters. With the exception of probing scores of the *Bias* tuned model, other par-tial finetuning methods have roughly the same probing scores compared to that full finetuning. This indicates that MechanisticProbe can provide consistent analy-sis for LMs finetuned in different ways.

The direct visualization of $\mathbb{E}[\pi(\boldsymbol{A}_{\text{simp}})]$ for dif-ferent finetuning methods can be found Figure 16(a-d). And the test accuracy of these 4 finetuned mod-els are: 99.42, 99.31, 94.11, and 76.84. We can find that finetuning attention parameters or MLP parameters can obtain quite similar attention pat-

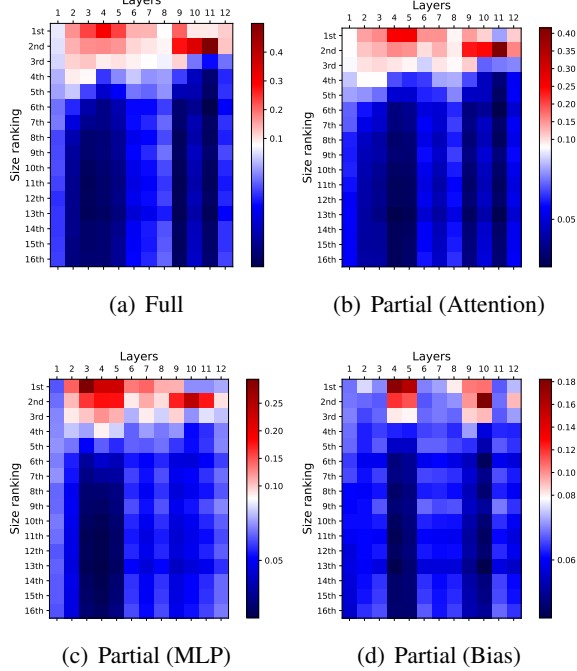

(a) Full  (b) Partial (Attention)

(c) Partial (MLP)  (d) Partial (Bias)

Figure 16: Visualization of $\mathbb{E}[\pi(\boldsymbol{A}_{\text{simp}})]$ with different ways of finetuning. The $x$-axis represents layers and the $y$-axis represents size ranking. We take mean pooling attention heads. From Figures (a-d) are GPT-2 with full finetuning, GPT-2 with partial finetuning on atten-tion parameters, GPT-2 with partial finetuning on MLP parameters, and GPT-2 with partial finetuning on bias parameters. We can find that different finetuning meth-ods would not affect the attention patterns much.

terns to that of full model finetuning. Regarding the bias tuning, it is slightly different. We speculate that this is because the LM does not learn the $k$-th smallest element task well (with much lower test accuracy). Generally, without significant perfor-mance drops, these results can support the general usage of our probing method MechanisticProbe under different finetuning settings.

### B.7 Task Difficulty & Model Capacity

This subsection discusses which factors can let GPT-2 handle reasoning tasks with more leaf nodes in $G$, i.e., large $k$. We explore the model capacity and reasoning task difficulty. Specifically, we ex-tend the list length $m$ from 16 to 64, and maximum $k$ from 8 to 32. For each $k$, we finetune LMs with the same finetuning settings and evaluate them on test data with accuracy. For model capacity, we compare three versions of GPT-2 with different sizes: Distilled GPT-2, GPT-2 (small), and GPT-2 Medium. For task difficulty, we construct synthetic data by selecting $m = 64$ numbers from 256, 384,

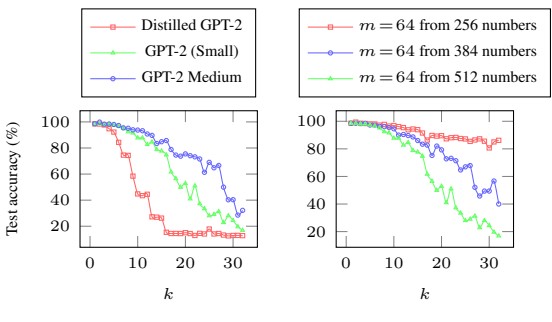

(a) Model capacity      (b) Procedural task difficulty

Figure 17: Test accuracies for LMs under different conditions. The $x$-axis represents $k$ (finding the $k$-th smallest number) and the $y$-axis represents the test accuracy. The left figure explores the performance of LMs with different capacities, and the right figure explores the performance of GPT-2 with different task difficulties.

and 512 distinct numbers.

We report their test accuracies in Figure 17. Note that here the setting remains the same: for each reasoning task (i.e., $k$), we finetune an individual model. From Figure 17(a), we find that LMs with large model capacities can better solve procedural tasks with more complex $G$ (i.e., more leaf nodes in $G$). But it does not mean that small LMs fail in this case. If we can reasonably reduce the task difficulty (e.g., decompose the procedural task), small LMs are still able to handle that task with complex $G$ (Figure 17(b)).

### B.8    What if Reasoning Tasks Become Harder?

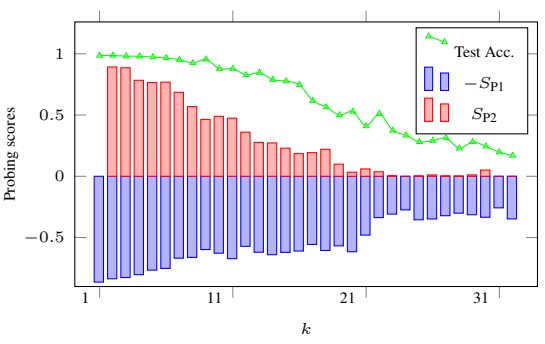

Figure 18: Probing scores and test accuracy of GPT-2$_{FT}$ on more difficult reasoning tasks. We finetune GPT-2 models to find the $k$-th smallest number from the long number list ($m = 64$, and $k$ ranges from 1 to 32). Results show that when the accuracy is low, GPT-2$_{FT}$ would still know how to select top-$k$ numbers more or less. But they are unable to find the $k$-th smallest number from top-$k$ numbers anymore.

Till now, we have experimented on a relatively easy task ($m = 16$). In this subsection, we increase the difficulty of the task by extending the input number list from $m = 16$ numbers to $m = 64$ numbers. We report the test accuracy as well as the two probing scores in Figure 18, varying $k$ from 1 to 32. As expected, the test accuracy decreases smoothly from near $100\%$ to around $15\%$. Interestingly, $S_{P1}$ and $S_{P2}$ do not decrease with the same speed. Even when the model has a very low accuracy, $S_{P1}$ still maintains at a high score (above $30\%$), while $S_{P2}$ quickly jumps to 0 when $k$ is around 20. This suggests that GPT-2$_{FT}$ solves the two steps sequentially, and step 2 fails first when the task goes beyond the capacity of the model.

## C    Supplementary about LLaMA

### C.1    Settings for 4-shot and Finetuned LLaMA

For the in-context learning of LLaMA, we construct the input prompt as simple as possible. Given set of statements $[S_1, S_2, ...]$, the question statement $Q$, and the answer label A (e.g., "True" or "False"), the prompt templates for ProofWriter and ARC are:

$$[S_1, S_2, ...] + [Q] + \text{True or False?} + [A],$$
$$[S_1, S_2, ...] + [Q] + \text{The answer is:} + [A].$$

The test accuracy of 0-shot, 2-shot, 4-shot, and 8-shot prompting of LLaMA can be found in Table 5. We select 4-shot in-context learning setting in our analysis due to its best performance.

Table 5: Test accuracy of LLaMA.

| ProofWriter | | | | |
|---|---|---|---|---|
| Acc. (%) | depth=0 | depth=1 | depth=2 | depth=3 |
| 0-shot LLaMA | 50.07 | 49.61 | 49.74 | 49.83 |
| 2-shot LLaMA | 75.58 | 75.83 | 74.31 | 73.74 |
| 4-shot LLaMA | 81.72 | 78.33 | 75.58 | 74.64 |
| 8-shot LLaMA | 54.26 | 52.96 | 52.82 | 52.42 |
| Finetuned LLaMA | 100 | 100 | 100 | 100 |
| ARC | | | | |
| 4-shot LLaMA | - | 56.32 | 53.40 | - |

Regarding the finetuning of LLaMA (i.e., partially finetuning on attention parameters), most settings are similar to that of GPT-2. The epoch number is set as 2, and the batch size is 256. We use the AdamW optimizer with weight decay $1e - 5$ for finetuning, and the warmup number is 500. The

learning rate is $1e-6$. Test accuracy of finetuned models can be found in Table 5 as well.

## C.2 Layer (Attention) Pruning

For the layer pruning, we use the greedy search strategy. Specifically, we remove all attentions in layers from top to bottom.[15] If the performance decrease on test data is small (less than $5\%$ in total), the attention in that layer is dropped. For 4-shot LLaMA on ProofWriter, 13 (out of 32) top layers are removed, and 15 (out of 32) top layers are removed for ARC. For finetuned LLaMA on ProofWriter, 2 middle layers (layer 9 and layer 13) and 16 top layers are removed. After removing all attentions in these layers, the performance decreases are around $2\%$ for both in-context learning and finetuning settings.

## C.3 Statistics of Cleaned ProofWriter and Annotated ARC

Table 6: Data statistics of cleaned ProofWriter in terms of different depth.

| # of examples | Training | Development | Test |
|---|---|---|---|
| depth $= 0$ | 84,568 | 12,227 | 24,270 |
| depth $= 1$ | 41,718 | 6,101 | 12,044 |
| depth $= 2$ | 25,021 | 3,712 | 7,215 |
| depth $= 3$ | 14,042 | 2,079 | 4,132 |
| depth $= 4$ | 6,078 | 891 | 1765 |
| depth $= 5$ | 5,998 | 874 | 1756 |

**ProofWriter.** We follow the original data split for training, development, and test sets. However, the depth split of ProofWriter is not suitable in our case. The original dataset only considers the largest depth of a set of examples (with similar templates) for the split. It means for example in depth 5, there would be many of them with depth smaller than 5. In our case, we classify examples into 6 types from depth 0 to 5 only based on the example's reasoning tree depth. After the depth split, we also remove examples whose reasoning trees have loops or multiple annotations. Besides, we remove few examples whose depth annotations are wrong (e.g., annotated as depth 5 but only with 4 nodes in $G$). Statistics of the cleaned and re-split ProofWriter

---

[15]Transformer layers without attention degrade to MLPs.

---

can be found in Table 6. We can find that there are less than 2000 4/5-depth examples in test data.

Table 7: Data statistics of annotated ARC in terms of different depth.

| # of examples | Training | Development | Test |
|---|---|---|---|
| depth $= 1$ | 331 | 51 | 87 |
| depth $= 2$ | 380 | 64 | 95 |
| depth $= 3$ | 277 | 28 | 67 |
| depth $= 4$ | 175 | 18 | 51 |
| depth $> 4$ | 150 | 26 | 40 |

**ARC.** We follow the original data split for training, development, and test sets (Ribeiro et al., 2023). Note that the number of examples in ARC is quite small. Thus, in our analysis, we do not run experiments only on test data. We simply merge all data for the analysis.

## C.4 Reasoning Tree Ambiguity Example

We consider a simple case to explore if the reasoning tree ambiguity issues happens in LLaMA. We sample 1024 examples whose annotations of 2-depth reasoning trees are

$$S_1 -> S_2 -> S3 \dashrightarrow Q.$$

We give a real example from the dataset randomly to illustrate the issue of reasoning tree ambiguity. Consider the three statements of $G$ (from 17 input statements) as

$S_1$ : Erin is cold;

$S_2$ : If someone is cold then they are rough;

$S_3$ : If someone is rough then they are white;

$Q$ : Erin is white (True).

It is intuitive that following the annotated reasoning tree could obtain correct answer. However, there are other ways to answer the question. We can first combine $S_2$ and $S_3$ to get a new statement $S_4$ as

$S_4$ : If someone is cold then they are white.

Then, the reasoning tree becomes

$$S_2 -> S_3 -> S_1 \dashrightarrow Q,$$

and we can rewrite it with brackets as

$$S_1 -> (S_2 -> S_3) \dashrightarrow Q.$$

There are multiple ways to do reasoning for this example, and we do not know which one the LM uses. Thus, in this work, we ignore these kinds of examples with reasoning tree depth larger than 1.[16]

## C.5 Original Probing Scores

Table 8: The original classification F1-Macro scores of LLaMA for two probing tasks.

| ProofWriter | | | | | | | |
|---|---|---|---|---|---|---|---|
| Probing task | | $S_{\text{F1}}(V\|\boldsymbol{A}_{\text{simp}})$ | | | $S_{\text{F1}}(G\|V, \boldsymbol{A}_{\text{simp}})$ | | |
| LLaMA | | random | 4-shot | finetuned | random | 4-shot | finetuned |
| depth $= 0$ | $\|\mathcal{S}\| = 1$ | 48.10 | 77.79 | 73.57 | - | - | - |
| depth $= 1$ | $\|\mathcal{S}\| = 2$ | - | - | - | 100 | 100 | 100 |
| | $\|\mathcal{S}\| = 4$ | 62.38 | 79.24 | 80.49 | 47.39 | 96.50 | 98.01 |
| | $\|\mathcal{S}\| = 8$ | 63.31 | 73.36 | 78.02 | 54.67 | 92.63 | 98.39 |
| | $\|\mathcal{S}\| = 12$ | 51.64 | 64.33 | 67.45 | 49.75 | 88.73 | 96.71 |
| | $\|\mathcal{S}\| = 16$ | 49.79 | 58.42 | 60.37 | 44.43 | 87.65 | 94.06 |
| | $\|\mathcal{S}\| = 20$ | 47.40 | 53.24 | 55.73 | 48.74 | 89.74 | 96.98 |
| | $\|\mathcal{S}\| = 24$ | 48.40 | 53.25 | 57.34 | 51.77 | 90.51 | 97.31 |
| ARC | | | | | | | |
| depth=1 | - | 51.21 | 98.77 | - | 48.73 | 80.38 | - |
| depth=2 | - | 50.86 | 98.28 | - | 50.38 | 76.88 | - |

The original classification scores (F1-Macro) for two probing tasks on LLaMA can be found in Table 8. Here, *random* means we randomly initialize LLaMA and use its attentions for probing as the random baseline.

---

[16]In ProofWriter, there are only 30% examples that have reasoning trees with depth larger than 1.