# OpenReview forum: "Towards a Mechanistic Interpretation of Multi-Step Reasoning Capabilities of Language Models"
_EMNLP/2023/Conference — EMNLP 2023 Main_

### Official Review · Reviewer_F7aK · 2023-07-27

**Soundness:** 4

**Excitement:**

4: Strong: This paper deepens the understanding of some phenomenon or lowers the barriers to an existing research direction.

**Paper Topic And Main Contributions:**

This work explores the internal multi-step reasoning process in LLMs. It introduces a novel probing approach called MechanisticProbe to recover the reasoning tree that resembles the correct reasoning process. The probe is used to analyze two LLMs (GPT-2 and LLaMA) on a synthetic task (finding the kth-smallest element) and a real-world reasoning task (ProofWriter) respectively. Using this probe, the authors find that the model’s attention patterns are indicative of the reasoning process represented by the reasoning tree. They also find that LLMs identify useful tokens/statements in the early layers and then reason over it step-by-step.

**Questions For The Authors:**

A. Can the authors provide some additional explanation/hypothesis as to why the model fails on longer inputs, despite being fine-tuned on them? Is this length beyond the sequence length that the model has been trained on?

B. In the kth-smallest element experiment, it is unclear if the gains in performance/attention patterns are learned exclusively during finetuning. Would finetuning a randomly initialized GPT-2 achieve similar results?

C. Can the authors expand more on the kNN classification of tokens/sentences as useful (similarly for the height classification)? Is the training data being used to identify the neighbors?

**Reasons To Accept:**

The paper is well-written and the experiments are carefully designed. With LLMs being prevalent across different fields and applications, it is important to understand their inner workings. This work sheds light on some interesting insights into the internals of model behavior that would be useful for the community to build upon.

**Reasons To Reject:**

Attributing model behavior to attention is not always reliable and it is unclear how this method can be scaled to more complex reasoning tasks. Additionally, correlation does not imply causation. As the authors have pointed out, just because attention encodes the correct reasoning tree, it does not mean that these trees are being used by the model, regardless of its correlation with the probing scores. I think a causal intervention approach would instill more confidence in believing that the model actually leverages the reasoning tree encoded in its attention for making predictions. Nevertheless, I believe the paper presents some interesting findings about models’ attention mechanisms.

**Reproducibility:**

3: Could reproduce the results with some difficulty. The settings of parameters are underspecified or subjectively determined; the training/evaluation data are not widely available.

**Reviewer Confidence:**

4: Quite sure. I tried to check the important points carefully. It's unlikely, though conceivable, that I missed something that should affect my ratings.

**Typos Grammar Style And Presentation Improvements:**

- Replace k-smallest with kth-smallest as k-smallest element confuses the reader with the top-k subtask.

- missing x-axis labels in Figure 9

- typos in lines 056, 209, 237

---

> ### Author Rebuttal · Authors · 2023-08-28
>
> *We thank the reviewer for recognizing the motivation and contribution of our work. Understanding the inner workings of LLMs when they do reasoning is very important at this time in the development of language models. To provide reliable insights, we carefully design experiments and try to be as comprehensive as possible. Concerns are responded to below point by point.*
>
> ---
>
> **Point 1: Scalability**
>
>     …and it is unclear how this method can be scaled to more complex reasoning tasks.
>
> **Response:**
> >Our method can be applied to many more complex reasoning tasks as long as they can be formalized as multiple-choice questions (e..g, MMLU, GSM8K), and the reasoning tree is chain-like (most of the reasoning examples that can be solved by current popular chain-of-thought prompting methods are chain-like.).
>
> ---
>
> **Point 2: Correlation does not imply causation.**
>
>     Attributing model behavior to attention is not always reliable…
>
>     Additionally, correlation does not imply causation. As the authors have pointed out, just because attention encodes the correct reasoning tree, it does not mean that these trees are being used by the model, regardless of its correlation with the probing scores. I think a causal intervention approach would instill more confidence in believing that the model actually leverages the reasoning tree encoded in its attention for making predictions.
>
> **Response:**
>
> >We thank the reviewer for the constructive comments. In section 4.3, we explore the relationship between our proposed probing scores and the LLM’s performance. Results imply that when we can successfully detect reasoning steps from LLM’s attention, the model is more likely to produce a correct prediction.
>
> >As for causal intervention, we classify attention heads into two groups: those focusing on number size (called size head) and others. The size heads contribute to the attention map for probing, while other heads can be regarded as noise in our analysis. We find that dropping size heads (less than 10% heads) has a much bigger impact than dropping other heads (>40% heads) (Figure 12). This supports the causation relationship between attention patterns and test accuracy.  We will also try to run experiments on ProofWriter and add the results as suggested.
>
> ---
>
> **Question A:**
>
>     Can the authors provide some additional explanation/hypothesis as to why the model fails on longer inputs, despite being fine-tuned on them? Is this length beyond the sequence length that the model has been trained on?
>
> **Response:**
> >The input length is fixed, it is k that is increased. We suppose that the model failed due to the increase in task difficulty. Even if we finetune LMs, limited model capacity could make it hard to learn such difficult tasks well. We will add discussions in section 3.4 to provide more intuitions.
>
> ---
>
> **Question B:**
>
>     In the kth-smallest element experiment, it is unclear if the gains in performance/attention patterns are learned exclusively during finetuning. Would finetuning a randomly initialized GPT-2 achieve similar results?
>
> **Response:**
> >No. We find that finetuning GPT-2 from scratch cannot converge even with millions of supervised data points (test accuracy < 1%). Note that the synthetic task for LMs is not that easy: LMs treat numbers as tokens, and they should first differentiate the size of numbers before giving the correct k-th smallest number. In Appendix B.3, we probe the pretrained GPT-2 (without any finetuning). We find that GPT-2 could learn to differentiate the size of numbers during pretraining.
>
> ---
>
> **Question C:**
>
>      Can the authors expand more on the kNN classification of tokens/sentences as useful (similarly for the height classification)? Is the training data being used to identify the neighbors?
>
> **Response:**
> >We run the KNN classifier under the cross-validation setting. Part of the data is used as training data. Then, during testing, the classifier would find the nearest neighbor of the test data and assign its neighbor’s label (useful / not useful) to it. We would add more descriptions about the KNN classifier in the paper for clarification.

---

### Official Review · Reviewer_3L4M · 2023-07-28

**Typos Grammar Style And Presentation Improvements:** Line 94-97
**Soundness:** 4

**Excitement:**

3: Ambivalent: It has merits (e.g., it reports state-of-the-art results, the idea is nice), but there are key weaknesses (e.g., it describes incremental work), and it can significantly benefit from another round of revision. However, I won't object to accepting it if my co-reviewers champion it.

**Paper Topic And Main Contributions:**

This paper analyzes multi-step reasoning of LLM by introducing a mechanistic probing task. LLMs perform well on multi-step reasoning tasks, but it is unclear whether they are truly reasoning or just using memorized content. This paper assumes that LLMs are indeed reasoning, and the reasoning progress can be expressed as a tree. Therefore, the paper presents a probing task that predicts the tree given the attention of the model. The task is further simplified and produces two scores that represent the likelihood of the tree. Experiments on GPT-2 and LLaMA suggest the existence of such trees. The LLaMA experiment also shows higher scores are correlated with better performance and robustness.

**Questions For The Authors:**

A. Is there a threshold for probing scores that indicate the tree exists? The S_p1 scores for the two experiments are quite different.

B. Do the two subtasks (sec. 2.3) represent a two-step solution to reasoning tasks? The first step is to select related statements, and the second step is to reach the conclusion given these statements.

**Reasons To Accept:**

1. The idea of discovering trees from the attention of LLMs is novel and timely.
2. This paper presents comprehensive experiments and analyses to verify such trees, and show their impact on performance and robustness.

**Reasons To Reject:**

1. The probing task lacks general applicability. It is based on several assumptions, like the task format (footnote 2), tree structure (line 147-149), and node dependence (footnote 5). Even for the two tasks presented, their problem setups are very different.
2. Given its small size and early training methods, GPT-2 may not qualify as a LLM that could do multi-step reasoning. The GPT-2 experiment, despite promising results, does not strongly support the claims.
3. How to compute F1-Macro scores (line 160-161) is not discussed.

**Reproducibility:**

3: Could reproduce the results with some difficulty. The settings of parameters are underspecified or subjectively determined; the training/evaluation data are not widely available.

**Reviewer Confidence:**

3: Pretty sure, but there's a chance I missed something. Although I have a good feel for this area in general, I did not carefully check the paper's details, e.g., the math, experimental design, or novelty.

---

> ### Author Rebuttal · Authors · 2023-08-28
>
> *We thank the reviewer for recognizing the novelty and contribution of our work. According to our knowledge, we are the first to propose an attention-based method for a mechanistic understanding of the LLM’s reasoning ability. We present comprehensive experiments verifying our analysis method and provide new insights for understanding the multi-step reasoning ability of LLMs. Responses to proposed concerns are given below point by point.*
>
> ---
>
> **Point 1:**
>
>     The probing task lacks general applicability. It is based on several assumptions, like the task format (footnote 2), tree structure (line 147-149), and node dependence (footnote 5). Even for the two tasks presented, their problem setups are very different.
>
> **Response:**
> >We presented two tasks that have the same task format: classification. For the synthetic task, each number is a label, and LMs do classification across all tokens in the vocabulary (for simplicity). For the ProofWriter, there are only two labels (true and false). LMs do classification across these two label tokens.
> >>1. *Task format:* A variety of reasoning tasks can be formalized as multiple-choice questions (e..g, MMLU, GSM8K), which can also be easily adapted into our format.
> >>2. *Tree structure:* Even if the reasoning structure is very complex, most of them in practice are chain-like. We consider reasoning structures as simple trees, which cover most reasoning examples in reality. Specifically, most the reasoning examples that can be solved by current popular chain-of-thought prompting methods are chain-like.
> >>3. *Node dependence:* This assumption is not necessary. We predict node depths independently because it suffices to recover the reasoning tree. Otherwise, there are many existing tools available [1] that consider the node dependence for extracting trees from the attention patterns. Given that our work is the first such attempt, we make several simplifying assumptions.
>
> [1] Hewitt, John, and Christopher D. Manning. "A structural probe for finding syntax in word representations." Proceedings of the 2019 Conference of the North American Chapter of the Association for Computational Linguistics: Human Language Technologies, Volume 1 (Long and Short Papers). 2019.
>
> ---
>
> **Point 2 and Point 3:**
>
>     Given its small size and early training methods, GPT-2 may not qualify as a LLM that could do multi-step reasoning. The GPT-2 experiment, despite promising results, does not strongly support the claims.
>     How to compute F1-Macro scores (line 160-161) is not discussed.
>
> **Response:**
> >We will change the description (LLM to LM) to avoid misunderstanding, and add more details about F1-Macro score computation in the paper. Our approach is agnostic to model size.
>
> ---
>
> **Question A:**
>
>     Is there a threshold for probing scores that indicate the tree exists?
>
> **Response:**
> >We use the random baseline as a threshold (i.e., S_p1/S_p2=0). We also computed the p-value using a permutation test. The p-value is always is 0, so we didn’t include it in the results.
>
> ---
>
> **Question B:**
>
>     Do the two subtasks (sec. 2.3) represent a two-step solution to reasoning tasks?...
>
> **Response:**
> >Yes. The first subtask/step selects the k leaves in the reasoning tree, and the second subtask/step selects the root from the leaves.

---

### Official Review · Reviewer_jwyS · 2023-08-04

**Typos Grammar Style And Presentation Improvements:** 1. The authors don't need to repeated…
**Soundness:** 4

**Excitement:**

4: Strong: This paper deepens the understanding of some phenomenon or lowers the barriers to an existing research direction.

**Missing References:**

* The authors should cite "Attention is not Explanation" and "Attention is not not Explanation"

**Paper Topic And Main Contributions:**

Summary: The paper investigates if large language models (LLMs) such as GPT-2 and LLaMA perform multi-step reasoning tasks by memorizing answers or through a procedural reasoning mechanism. The authors propose a new probing approach, MechanisticProbe, that recovers reasoning trees from the model's attention patterns for a given problem. They test the probe on a synthetic task (k-smallest element) and a real-world task (ProofWriter) and find that reasoning trees can be largely recovered, suggesting that LLMs go through multi-step reasoning within their architecture. Higher probing scores correlate with better performance and noise tolerance, highlighting the importance of capturing the correct reasoning process.

Contributions:
1. The proposal of a new probing approach, MechanisticProbe, which recovers reasoning trees from LLMs' attention patterns, helping to gain insights into the multi-step reasoning process of these models.
2. The successful application of MechanisticProbe on GPT-2 and LLaMA, on synthetic and real-world tasks, providing evidence to suggest that LLMs indeed go through a multi-step reasoning process within their architecture.
3. The discovery that higher probing scores correlate with better model performance and robustness against noise, implying the importance of capturing the correct reasoning process for LLM accuracy and robustness.

**Questions For The Authors:**

1. It's strange that one model is only evaluated on one dataset.  Is there a particular reason that makes the authors do this?
2. It's extremely misleading that the authors call proofwriter a real world task, which really isn't.  Proofwriter is synthetically generated and not even in purely natural language.
3. I'm skeptical of how well this works in reality.  Based on the evidence shown in the two papers "Attention is not Explanation" and "Attention is not not Explanation", attention cannot be served as explanation, especially for real language datasets.  The authors should consider add hotpotqa and musique, which are procedural reasoning datasets and in natural language.

**Reasons To Accept:**

1. The problem studied is interesting and important.
2. Some interesting analysis has been done.

**Reasons To Reject:**

1. The experiments are limited, w.r.t. both datasets and models tested.  For datasets, k-th smallest number task is too simplified, and proofwriter is a totally synthetically generated dataset.  And only one model is tested for one dataset.
2. Although the authors demonstrate that attention patterns can help recover reasoning trees, the interpretability of attention patterns is a subject of ongoing debate. (See questions) The probing approach's reliance on attention patterns may raise concerns about its ability to provide a definitive mechanism to interpret LLMs.
3. Unclear how this method scales to the actual large models people use these days

**Reproducibility:**

2: Would be hard pressed to reproduce the results. The contribution depends on data that are simply not available outside the author's institution or consortium; not enough details are provided.

**Reviewer Confidence:**

4: Quite sure. I tried to check the important points carefully. It's unlikely, though conceivable, that I missed something that should affect my ratings.

---

> ### Author Rebuttal · Authors · 2023-08-28
>
> *We thank the reviewer for recognizing the motivation of our work. Targeting an interesting question, we propose a new attention-based analysis method for LLMs, and provide new insights about how LLMs procedurally perform multi-step reasoning step-by-step within their architecture. However, we would like to respectfully point out that there may be a few misunderstandings about our method and experiment design. Responses are given below point by point.*
>
> ---
>
> **Point 1 & Question 1: The experiment setting is a bit strange.**
>
>     The experiments are limited, … And only one model is tested for one dataset.
>     It's strange that one model is only evaluated on one dataset. Is there a particular reason that makes the authors do this?
>
> **Response:**
> >Regarding the k-th smallest number task, we did experiments on other models with different model capacities: GPT-2 medium, and Distilled GPT-2 (see Appendix B.5). And similar results were obtained. We will add the results of LLaMA in the camera-ready version.
>
> >For the ProofWriter task, in order to obtain meaningful analysis, the LLMs need to have a reasonable performance on the task. At the same time, due to the limitation of computing resources, we can only afford to conduct our experiment with LLMs with moderate numbers of parameters.
> >We have experimented on GPT-J (6B) and GPT-Neo (2.7B). However, their test accuracies are \~50% (GPT-J w/ in-context learning) and \~60% (GPT-J w/ finetuning); \~50% (GPT-Neo w/ in-context learning) and \~60% (GPT-Neo w/ finetuning), which is only slightly better than random guess (\~50%). LLaMA 7B is the smallest model that can achieve >80% test accuracy under the in-context learning setting (see Table 2 in the paper).
>
> ---
>
>
> **Point 1 & Question 2 & Question 3: Experiment datasets are not real-world data.**
>
>     For datasets, k-th smallest number task is too simplified, and proofwriter is a totally synthetically generated dataset.
>     It's extremely misleading that the authors call proofwriter a real world task, which really isn't. Proofwriter is synthetically generated and not even in purely natural language.
>     The authors should consider add hotpotqa and musique, which are procedural reasoning datasets and in natural language.
>
> **Response:**
> >In our experiments, we first proposed a synthetic task to give readers preliminary insights into how LLMs perform reasoning. Then, we introduced a more practical reasoning task. We think ProofWriter is a suitable task for our analysis for the following reasons:
> >>1. We agree that “real-world task” might not be accurate for ProofWriter and one could argue that the sentences are not “natural” enough, we will rephrase this in the paper to avoid misunderstanding. However, the task still requires logical reasoning and an understanding of certain concepts in human language.
> For example, *“Charlie is blue; If someone is blue then they are kind -> Charlie is kind.”*. The LLMs need to know that Charlie falls within the scope of “someone” and understand the if-then statement.
> 2. Reasoning analysis on synthetic data (i.e., ProofWriter) is commonplace and sometimes desirable because it allows us to study some phenomena not easy to do in natural text [2]. For instance, ProofWriter comes with the annotation of reasoning trees, which provides a ground truth to be compared with the extracted tree from LLMs. The lack of such reasoning trees in many real-world datasets (e.g., HotpotQA) makes it inadequate for such analysis.
> 3. Various real-world datasets could have already been used for pretraining LLMs (see Tables 1&9 in [1]). For example, both HotpotQA and MuSiQue are based on Wikipedia. This could cause data leakage and LLMs would learn shortcuts to solve these multi-step reasoning tasks [3]. This makes them unsuitable to be used to answer our main research questions.
>
> [1] Longpre, Shayne, et al. "A Pretrainer's Guide to Training Data: Measuring the Effects of Data Age, Domain Coverage, Quality, & Toxicity." arXiv preprint arXiv:2305.13169 (2023).
> [2] Saparov, Abulhair, and He He. "Language models are greedy reasoners: A systematic formal analysis of chain-of-thought." arXiv preprint arXiv:2210.01240 (2022).
> [3] Trivedi, Harsh, et al. "MuSiQue: Multihop Questions via Single-hop Question Composition." Transactions of the Association for Computational Linguistics 10 (2022): 539-554.
>
> ---
>
> **Point 2 & Question 3: Using attention for interpretation could be problematic.**
>
>     The probing approach's reliance on attention patterns may raise concerns about its ability to provide a definitive mechanism to interpret LLMs.
>     I'm skeptical of how well this works in reality. Based on the evidence shown in the two papers "Attention is not Explanation" and "Attention is not not Explanation", attention cannot be served as explanation, especially for real language datasets.
>
> **Response:**
> >Thanks for the references, we will add them in the paper. First of all, we would like to respectfully point out that the two references only discussed the attention mechanism of RNN/LSTM. Analyzing transformers (“attention is all you need”) via attention is popular and many attention-based interpretation methods/findings have been proposed in recent years.
>
> >We understand the concern that the attention mechanism could fail in model interpretation. However, as said in [4], we do not think [5] disproves the usefulness of attention mechanisms for explainability. More specifically, [5] claims “attention distributions often have no effect on model outputs, while we include two experiments (Section 4.3 & Appendix B.4) to verify that the attention patterns do signify the model’s reasoning ability:
> >>1. Section 4.3. We explore the relationships between our proposed probing scores and an LLM’s performance. Results imply that when we can successfully detect reasoning steps of useful statements from LLM’s attention, the model is more likely to produce a correct prediction.
> 2. Appendix B.4. We classify attention heads into two groups: those focusing on number size (called size head) and others. The size heads contribute to the attention map for probing, while other heads can be regarded as noise for our analysis. We find that dropping size heads (less than 10% heads) has a much bigger impact than dropping other heads (>40% heads) (Figure 12). This supports the causation relationship between attention patterns and test accuracy.
>
> [4] Wiegreffe, Sarah, and Yuval Pinter. "Attention is not not explanation." arXiv preprint arXiv:1908.04626 (2019).
> [5] Jain, Sarthak, and Byron C. Wallace. "Attention is not explanation." arXiv preprint arXiv:1902.10186 (2019).
>
> ---
>
> **Point 3: Scalability.**
>
>     Unclear how this method scales to the actual large models people use these days
>
> **Response:**
> >The models we experiment on have a wide range of applications in real-world scenarios. Both GPT-2 and LLaMA have been widely used in both academia and industry.

---

### Meta-Review · Area_Chair_CSQj · 2023-09-18

**Recommendation:** 4

**Metareview:**

The reviewers were generally on the same page that the current scope of the work is somewhat narrow (on a synthetic task and on the naturalistic but synthetically-generated ProofWriter task), yet the overall findings about being able to mechanistically recover the implicit tree-structured reasoning from the model is valuable enough to be of interest. Unlike most explainability work that tries to obtain natural language multi-step explanations, the mechanistic interpretability angle of this work is novel in the multi-step reasoning space and provides useful insights.

The authors adequately addressed many of the concerns raised during the review period. The findings seem valuable to the community, even if currently somewhat narrow in scope. The reviewers have suggested specific other datasets (e.g., Musique) to push this line of work further.

---

### Decision · Program_Chairs · 2023-10-07

**Decision:**

Accept-Main

**Comment:**

The reviewers were generally on the same page that the current scope of the work is somewhat narrow (on a synthetic task and on the naturalistic but synthetically-generated ProofWriter task), yet the overall findings about being able to mechanistically recover the implicit tree-structured reasoning from the model is valuable enough to be of interest. Unlike most explainability work that tries to obtain natural language multi-step explanations, the mechanistic interpretability angle of this work is novel in the multi-step reasoning space and provides useful insights.

The authors adequately addressed many of the concerns raised during the review period. The findings seem valuable to the community, even if currently somewhat narrow in scope. The reviewers have suggested specific other datasets (e.g., Musique) to push this line of work further.